# Loss of *MXRA8* Delays Mammary Tumor Development and Impairs Metastasis

**DOI:** 10.3390/ijms241813730

**Published:** 2023-09-06

**Authors:** Kaitlyn E. Simpson, Christina A. Staikos, Katrina L. Watson, Roger A. Moorehead

**Affiliations:** Department of Biomedical Sciences, Ontario Veterinary College, University of Guelph, Guelph, ON N1G 2W1, Canada; ksimps07@uoguelph.ca (K.E.S.); cstaikos@uoguelph.ca (C.A.S.); kawatson@uoguelph.ca (K.L.W.)

**Keywords:** TNBC, *MXRA8*, metastasis, tumor initiation, xenograft model

## Abstract

Matrix-remodeling-associated protein 8 or MXRA8 is a transmembrane protein that can bind arthritogenic alpha viruses like the Chikungunya virus and provide viral entry into cells. MXRA8 can also interact with integrin β3 and thus possibly regulate cell–cell interactions and binding to the extracellular matrix. While *MXRA8* has been associated with reduced survival in patients with colorectal and renal clear cell cancers, the role of *MXRA8* in breast cancer remains largely unexplored. Therefore, the aim of this research was to determine the role of *MXRA8* in breast cancer by knocking out *MXRA8* in the human triple-negative breast cancer cell line MDA-MB-231. The loss of *MXRA8* reduced cell proliferation in vitro but had no effect on apoptosis or migration in cultured cells. However, the loss of *MXRA8* significantly delayed tumor development and reduced metastatic dissemination to the lungs in a xenograft model. RNA sequencing identified three genes, *ADMATS1*, *TIE1*, and *BMP2*, whose expression were significantly reduced in *MXRA8*-knockout tumors compared to control tumors. *MXRA8* staining of a human breast cancer tissue array revealed higher levels of *MXRA8* in primary tumors and metastases of aggressive tumor subtypes (TNBC and HER2^+^) compared to less aggressive, ER^+^ breast cancers. Our findings demonstrate for the first time that *MXRA8* regulates the progression of human TNBC possibly through influencing the interaction of tumor cells with their microenvironment.

## 1. Introduction

Matrix-remodeling-associated protein 8 (MXRA8) (also known as limitrin, DICAM, and ASP3) is a transmembrane protein that was first identified in glial cells in 2003 [1]. Since its discovery, most of the MXRA8 research has focused on MXRA8′s function as a transmembrane protein that can bind arthritogenic alpha viruses like the Chikungunya virus and provide viral entry into cells [2]. Other functions attributed to MXRA8 include regulation of the blood–brain barrier, cell–cell adhesion, and integrin signaling specifically through heterophilic interaction with integrin αVβ3 [3,4,5]. 

The characterization of *MXRA8* in most cancers, including breast cancer, is limited. *MXRA8* is frequently expressed at high levels in solid tumors compared to adjacent normal tissue [6]. In colorectal cancer, *MXRA8* was associated with a poor prognosis and correlated with metastasis, recurrence, and an immunosuppressive tumor microenvironment [7]. *MXRA8* has also been associated with reduced survival in renal clear cell carcinoma [8], and *MXRA8* promotes glioma by regulating ferroptosis [9]. *MXRA8* is part of a 13-gene panel associated with reduced survival in esophageal squamous cell carcinoma [10], a 10-gene interaction network in osteoblastic sarcoma [11], and a 7-gene signature associated with metastasis in thyroid cancer [12]. However, these studies are correlative in nature and do not investigate the impact of manipulating *MXRA8* levels in cancer.

Our previous work implicated *MXRA8* in the growth and metastasis of triple-negative breast cancer (TNBC) [13]. TNBC is highly aggressive and lacks estrogen receptors (ERs) and human epidermal growth factor receptor 2 (HER2). While investigating potential therapeutic strategies for treating TNBC, our lab found that TNBC expresses very low levels of the microRNA family miR-200, and the re-expression of miR-200 in the TNBC cell line MDA-MB-231 inhibited tumor growth and metastasis in vivo [13]. RNA sequencing of the control and miR-200 re-expressing MDA-MB-231 cells identified *MXRA8* as one of the most significantly downregulated genes in MDA-MB-231 cells re-expressing the miR-200c/141 cluster. The re-expression of the miR-200c/141 cluster in MDA-MB-231 cells reduced *MXRA8* expression as well as tumor growth and metastasis in vivo. MXRA8 was also expressed at high levels in spontaneous lung metastases of MDA-MB-231 tumors [13].

To evaluate the function of *MXRA8* in TNBC, the current study knocked out *MXRA8* in MDA-MB-231 cells using CRISPR-Cas9. Loss of *MXRA8* expression delayed tumor development and metastasis in a xenograft model, thus providing the first evidence that reducing *MXRA8* can suppress TNBC metastasis. *MXRA8* appears to regulate these processes by altering genes like *ADAMTS1* and *TIE1* that influence the tumor microenvironment. Moreover, breast cancers from patients with aggressive subtypes (i.e., TNBC and HER2^+^) expressed higher levels of MXRA8 protein in primary tumors and metastatic lesions compared to less aggressive, ER^+^ breast cancers.

## 2. Results

### 2.1. Characterization of MXRA8-Knockout Clones In Vitro

Two independent *MXRA8*-knockout clones (231MXRA8KO-O and 232MXRA8KO-V) as well as a control clone (231MXRA8Con) were further characterized in vitro. The two *MXRA8*-knockout clones were selected as 231MXRA8KO-O cells had an 18-base deletion in *MXRA8* (based on DNA sequencing, Appendix A) and very low *MXRA8* expression (Figure 1A), while 231MXRA8KO-V had the largest *MXRA8* deletion (100 bases, based on DNA sequencing, Appendix A) and very low *MXRA8* expression (Figure 1A). MXRA8 expression was reduced by 97% in 231MXRA8KO-O and 98% in 231MXRA8KO-V cells compared to 231MXRA8Con cells.

Proliferation was assessed using BrdU incorporation and flow cytometry. As shown in Figure 1B, 231MXRA8KO-O cells had significantly lower proliferation than 231MXRA8Con cells. Proliferation rate was reduced by ~16% in 231MXRA8KO-V cells compared to 231MXRA8Con cells, but this value was not significant (Figure 1B). Basal apoptotic rates (Figure 1C) and transwell migration assays were also analyzed, but there were no significant differences between MXRA8-knockout cells and control cells in either of these assays.

As MXRA8 might mediate some of the actions of miR-200s, and miR-200s have been shown to increase the expression of the epithelial gene *CDH1* while lowering the expression of mesenchymal genes such as *ZEB1*/*ZEB2*, *TWIST1*/*TWIST2*, and *SNAI1*/*SNAI2*, the expression of these genes was evaluated (Figure 1C–J). Only *ZEB2* (Figure 1J) was significantly reduced in both 231MXRA8KO-O and 231MXRA8KO-V cells compared to 231MXRA8Con cells. None of the other epithelial or mesenchymal genes differed significantly.

RNA sequencing was performed on three independent RNA samples from 231MXRA8Con, 231MXRA8KO-O, and 231MXRA8KO-V cells. Aligned reads in the *MXRA8* gene were visualized in IGV, and Appendix A shows that all 231MXRA8KO-O and 231MXRA8KO-V clones had a region that was not mapped by any reads, and this is the region our *MXRA8* guide RNA was reported to target (*MXRA8* gRNA targeted hg38:chr1:1354815-1354837). The region the *MXRA8* gRNA targeted is found in all four protein-coding transcripts of the *MXRA8* gene and thus should induce a mutation in all *MXRA8* protein-coding transcripts. Table 1 shows the TPM values for the three protein-coding *MXRA8* transcripts that had TPM values > 1. For each transcript, the levels were reduced to zero or nearly zero in both *MXRA8*-knockout clones.

### 2.2. Characterization of 231MXRA8KO Clones In Vivo

To determine whether the loss of *MXRA8* expression regulated mammary tumor initiation, growth, or metastasis, 1 × 10^6^ 231MXRA8Con, 231MXRA8KO-O, or 231MXRA8KO-V cells were injected into the fourth mammary gland of NCG mice (NOD-*Prkdc^em26Cd52^Il2rg^em26Cd22^*/NjuCrl, Charles River, Wilmington MA, USA). Two different sets of injections were performed. In the first set of injections, 231MXRA8Con, 231MXRA8KO-O, and 231MXRA8KO-V cells were injected into three mice each. These mice were used as a pilot study to determine if the loss of *MXRA8* impacted mammary tumor development in vivo. Mammary tumor onset was assessed via palpation, and once a tumor developed, it was measured twice per week using digital calipers. As shown in Figure 2A, tumors induced by the injection of 231MXR8Con cells produced palpable tumors more quickly than either 231MXRA8KO-O or 231MXRA8KO-V tumors, and the 231MXRA8Con tumors grew rapidly to approximately 10% of body weight by 44 days post-injection.

A second set of injections was performed with seven mice being used for each cell line. One mouse injected with 231MXRA8KO-O cells became sick and had to be removed from the study, and thus, only tumor onset was evaluated in this mouse. Figure 2A–C shows tumor onset was significantly delayed in both 231MXRA8KO-O and 231MXRA8KO-V injections compared to 231MXRA8Con injections (the data in Figure 2C were derived from both sets of injections). Tumors produced by the injection of 231MXRA8KO-O cells grew significantly slower than tumors produced by 231MXRA8Con injections (Figure 2D), and 231MXRA8KO-V tumors grew at a similar rate as control tumors (Figure 2D). With this set of mice, tumors were collected when they reached approximately 10% of the mouse’s body weight. Thus, the 231MXRA8Con tumors were collected 44 days post-injection (n = 10 over both sets of injections), the 231MXRA8KO-V tumors were collected 65 days post-injection (n = 7), and the 231MXRA8KO-O tumors were collected 90 days post-injection (n = 6). Average tumor size at the time of collection was 682 ± 51 mm^3^ for 231MXRA8KOCon, 588 ± 93 mm^3^ for 231MXRA8KO-O, and 856 ± 76 mm^3^ for 231MXRA8KO-V tumors (Figure 2E). There was no statistically significant difference in the tumor size of either MXRA8-knockout tumors compared to control tumors at the time of collection. Histologically, tumors induced by 231MXRA8Con (Figure 2F,G) and 231MXRA8KO-V (Figure 2H,K) cells were composed primarily of cuboidal-shaped epithelial cells and frequently contained large regions of necrosis. 231MXRA8KO-O tumors typically had smaller regions of necrosis and more variability in tumor cell shape (Figure 2G,I).

Lung tissue was evaluated histologically and following vimentin immunohistochemistry. We have previously shown that a human-specific, anti-vimentin antibody stains metastatic MDA-MB-231 cells but not murine lung cells [14]. Vimentin expression, as determined by RNA sequencing, was not significantly different in 231MXRA8Con, 231MXRA8KO-O, and MXRA8KO-V tumors. The percentage of lung area occupied by vimentin-positive cells was determined using QuPath software v0.4.3 [15], and mice injected with 231MXRA8Con cells (n = 10, both sets of injections) had a significantly higher lung metastatic burden than mice injected with 231MXRA8KO-O (n = 6, second set of injections only) or 231MXRA8KO-V (n = 7, second set of injections only) cells (Figure 3A–G). Since all primary mammary tumors were approximately the same size when they were collected, and mice bearing 231MXRA8KO-O and 231MXRA8KO-V tumors had more time to metastasize (90 and 65 days, respectively) compared to 231MXRA8Con tumors (44 days), this observation truly represents a decrease in lung metastases in *MXRA8*-knockout tumors.

To evaluate how *MXRA8* regulates mammary tumor onset and metastasis, RNA sequencing was performed on three independent samples of each tumor induced by the injection of 231MXRA8Con, 231MXRA8KO-O, or 231MXRA8KO-V cells. Hierarchical clustering (Figure 4A) using a Euclidean distance measure and PCA analysis (Figure 4B) revealed that the 231MXRA8KO-O and 231MXRA8KO-V tumors were more similar to each other than 231MXRA8Con tumors.

Using a log fold change (FC) > 1 and a false discovery rate (FDR) < 0.01, 231MXRA8KO-O tumors expressed 1041 genes at higher levels and 1434 genes at lower levels than control tumors. The 231MXRA8KO-V tumors expressed 460 genes at higher levels and 1228 genes at lower levels compared to control tumors. The 231MXRA8KO-O and 231MXRA8KO-V tumors shared 1092 genes that were differentially expressed compared to the control tumors. The top ENCODE and ChEA pathway in both the 231MXRA8KO-O vs. 231MXRA8Con tumor and the 231MXRA8KO-V vs. 231MXRA8Con tumor comparisons was SUZ12 (Figure 4C). Similarly, the top biological process and cellular component were related to the extracellular matrix, cell adhesion, and the plasma membrane (Figure 4C). When the differentially expressed genes were ranked based on FDR, 11 genes were found in the top-25 genes of both the 231MXRA8KO-O vs. 231MXRA8Con tumors and the 231MXRA8KO-V vs. 231MXRA8Con tumors, and these genes are shown in the heatmap in Figure 4D. In particular, *ADAMTS1*, *TIE1*, and *BMP2* were in the top-10 differentially expressed genes in both 231MXRA8KO-O vs. 231MXRA8Con tumor and 231MXRA8KO-V vs. 231MXRA8Con tumor comparisons.

### 2.3. MXRA8 Protein Levels in Human Breast Cancer

To determine whether *MXRA8* was relevant in human breast cancer, a tissue microarray was probed with an anti-MXRA8 antibody. Figure 5A–C shows representative MXRA8 staining in an ER^+^ tumor, a TNBC, and a HER2^+^ tumor. Although the amount of MXRA8 staining was variable across primary tumors and metastatic lesions, aggressive tumors like TNBC and HER2^+^ had significantly higher levels of MXRA8 staining in primary tumors and metastatic lesions compared to less aggressive, ER^+^ tumors (Figure 5D,E).

## 3. Discussion

Our previous work identified *MXRA8* as one of the genes regulated by the miR-200 family that appeared to play a role in mammary tumorigenesis [13]. This finding was significant as only one other study had evaluated *MXRA8* in breast cancer, and this study reported that *MXRA8* was expressed in the stroma of breast and colorectal cancers [15]. However, no additional characterization of *MXRA8* was performed. Thus, our study is the first to manipulate *MXRA8* expression and determine its function in breast cancer. Our findings indicate that the loss of *MXRA8* expression suppresses mammary tumor onset and metastasis. Moreover, we found higher levels of MXRA8 protein in patient samples from aggressive breast cancer subtypes (TNBC and HER2^+^) compared to the less aggressive, ER^+^ subtype.

Exactly how *MXRA8* influences mammary tumor onset and metastasis remains unclear. The pathway analysis of differentially expressed genes in control and *MXRA8*-knockout tumors suggests that the loss of *MXRA8* influences the interaction between tumor cells and the tumor microenvironment. Pathways like extracellular matrix organization, cell–cell adhesion via plasma membrane, and collagen-containing extracellular matrix were identified. The ability of MXRA8 to influence the interactions between cells and the ECM is consistent with reports that MXRA8 can bind to β3 integrin [16]. Integrins are cell-surface receptors and are the main adhesion receptors that allow cells to interact with components of the ECM. There are 18 alpha and 8 beta subunits forming 24 heterodimers, and the heterodimers vary in their ability to bind specific proteins of the ECM. Integrins can also induce intracellular signaling via focal adhesion kinase leading to activation of the RAS/MAPK and PI3K/AKT signaling pathways [17]. Integrin heterodimers, including those containing β3 integrin have been implicated in regulating breast cancer metastasis [17,18,19,20].

The loss of *MXRA8* also led to a decrease in the expression of genes associated with breast cancer, including *ADAMTS1*, *TIE1*, and *BMP2*. ADAMTS1 is a member of the ADAMTS (a disintegrin and metalloproteinase with thrombospondin motifs) family of proteases that are secreted and bind to the extracellular matrix (ECM) [21]. ADAMTS1 is a protease that can modulate the ECM by cleaving its components, and it has been implicated in various physiological and pathological processes, including angiogenesis, inflammation, and tissue remodeling [21,22,23,24,25]. While an early study found that *ADAMTS1* was one of several *ADAMTS* genes downregulated in breast cancer compared to non-neoplastic breast tissue [26], more recent studies have associated high *ADAMTS1* expression with increased breast cancer metastasis. Lu et al. showed that the knockdown of *ADAMTS1* in MDA-MB-231 cells resulted in a reduction in metastatic burden, while the overexpression of *ADAMTS1* in the weakly metastatic MDA-MB-231 subline 2279 promoted bone metastases [27]. Similarly, Liu et al. [28] showed that the overexpression of *ADAMTS1* promoted metastasis of the mammary carcinoma cell line TA3, while a proteinase-dead ADAMTS1 prevented metastasis. Administration of an anti-ADMATS1 antibody reduced the growth of the murine mammary tumor cell line 4T1 in vivo [29]. In transgenic MMTV-PyMT mice, whole-body *ADAMTS1* knockout reduced mammary tumor size and lung metastatic burden [30].

Tyrosine kinase with immunoglobulin-like and EGF-like domains 1 (TIE1) is a receptor tyrosine kinase that is important in regulating endothelial growth and survival as well as lymphatic development [31,32,33]. While TIE2 can bind angiopoietin growth factors, ligands for TIE1 have yet to be identified [34]. TIE1 can be expressed in the endothelial cells of tumor blood vessels, and the loss of murine *Tie1* has reduced xenograft tumor growth and angiogenesis [32]. A high *TIE1* expression predicts poor outcomes in breast cancer patients [35]. 

Bone morphogenetic protein 2 (BMP2) is part of the transforming growth factor β superfamily, and there are over 20 different BMPs [36]. Several cell types of the mammary gland including epithelial cells and fibroblasts produce BMP2 to induce the proliferation of luminal progenitor cells [37,38]. High levels of BMP2 have been found in the tumor microenvironment of breast cancers and may promote epithelial transformation [37]. In xenograft models, BMP2 can promote EMT and metastasis [39], and BMP2 has been associated with poor disease-free survival in breast cancer patients [40].

Additional support of our findings stems from a study characterizing sublines of MDA-MB-231 cells with different metastatic potential. This study found that *ADAMTS1*, *TIE1*, and *BMP2* were all expressed at significantly higher levels in the highly metastatic sublines compared to a weakly metastatic subline [41]. 

Therefore, the loss of MXRA8 may influence intracellular signaling pathways that in turn alter the expression of genes like *ADAMTS1*, *TIE1*, and *BMP2* that regulate breast cancer metastasis. Alternatively, MXRA8-mediated signaling may regulate gene expression through histone methylation. SUZ12 ChEA was the top ENCODE and ChEA pathway in both MXRA8-knockout tumors compared to control tumors. SUZ12 is a component of the polycomb repressor complex 2 (PRC2) which methylates lysine 27 on histone H3. This histone methylation induces DNA condensation that reduces the access of gene promoters to transcription factors. Interestingly, *ADAMTS1* and *BMP2* are predicted to be regulated by SUZ12 [42]. It is also interesting to note that SUZ12 ChEA was also the top ENCODE and ChEA pathway in MDA-231c141 tumors, the tumors where we identified MXRA8′s potential role in breast cancer [13]. Therefore, it is possible that increased miR-200c/141 levels reduce *MXRA8* expression which in turn decreases the expression of breast cancer progression genes like *ADAMTS1* and *BMP2* potentially through increasing histone methylation. *MXRA8* (logFC = −7.9, FDR = 2.5 × 10^−209^), *ADAMTS1* (logFC = −3.7, FDR = 2.3 × 10^−14^), and *BMP2* (logFC = −2.9, FDR = 3.5 × 10^−8^) were all significantly downregulated in MDA-231c141 tumors compared to MDA-231EV tumors [13]. However, additional studies are required to further evaluate how *MXRA8* can influence breast cancer gene expression.

In summary, the loss of MXRA8 reduced breast cancer initiation and metastasis in human triple-negative breast cancer cells, and high levels of MXRA8 protein were associated with more aggressive breast cancer subtypes. These findings provide the first evidence that MXRA8 is an important regulator of breast tumorigenesis and the foundation for more detailed analysis of MXRA8′s role in breast cancer progression.

## 4. Materials and Methods

### 4.1. MXRA8 Knockout, Selection, and Cell Culture Conditions

MDA-MB-231 (cat # HTB-26) cells were purchased from ATCC (Manassas, VA, USA). To knock out MXRA8, MDA-MB-231 cells were initially infected with a hEF1a Blast-Cas9 lentiviral vector (cat# VCAS10126, Horizon/Dharmacon, Cambridge, UK). MDA-231Cas9 cells were selected with 20 µg/mL of blasticidin, and Cas9 protein was confirmed in the clones using Western blotting and an anti-Cas9 antibody (Appendix A, cat# 14697, New England Biolabs, Whitby, ON, Canada). MDA-231Cas9 cells were transduced with lentivirus that contained *MXRA8* gRNA targeting hg38:chr1:1354815-1354837 of *MXRA8* (VSGH10142-246490800, Horizon/Dharmacon, Cambridge, UK) or a control guide RNA (cat# VSGC10215, Horizon/Dharmacon, Cambridge, UK). Cells underwent selection using puromycin (ant-pr-1, InvivoGen, San Diego, CA, USA), and single-cell clones were seeded in 96-well plates and expanded. Genomic DNA was isolated from individual clones, and the DNA spanning the cleavage site underwent PCR and T7E1 mismatch detection (Horizon/Dharmacon, Cambridge, UK). DNA from clones with suspected *MXRA8* deletion was amplified using the MXRA8F1 primer GGC GTC AGG TAC CAG CAA GA and the MXRA8R1 primer CAC GTG GAG GAG GCT CAA CA followed by sequencing (Laboratory Services, University of Guelph, Guelph, ON, Canada). Clones with confirmed MXRA8 deletions were evaluated for *MXRA8* expression using *MXRA8* primers (unique assay ID: qHsaCED0045992, Bio-Rad Laboratories, Mississauga, ON, Canada) and qRT-PCR. Clones V (231MXRA8KO-V) and O (231MXRA8KO-O) were selected due to the lowest MXRA8 expression compared to MDA-MB-231 cells containing control gRNA (231MXRA8Con). 231MXRA8Con and knockout clones were maintained in DMEM media (GIBCO, Burlington, ON, Canada) supplemented with 10% FBS, 2% glutamine, 1% sodium pyruvate, 1% 4-(2-hydroxyethyl)-1-piper-azineethanesulfonic acid (HEPES), and 1% antibiotic/antimycotic. Cells cultures were maintained in media containing 1 µL/mL puromycin (ant-pr-1, InvivoGen, San Diego, CA, USA) and 2 µL/mL blasticidin (sc-495389, Santa Cruz Biotechnology, Dallas, TX, USA) for selection.

### 4.2. BrdU and Annexin V Flow Cytometry

For BrdU flow cytometry, an APC BrdU flow kit (BD Biosciences, San Jose, CA, USA, cat #552598) was used following the manufacturer’s protocol. Briefly, cells were incubated with 1mM BrdU in fully supplemented media for 24 h. Cells were then fixed, washed, and analyzed on an Accuri C6 cytometer (BD Biosciences, San Jose, CA, USA) using a flow rate of 35 µL/min, and 50,000 events were collected.

For annexin V flow cytometry, the BD Pharminogen APC Annexin V kit (BD Biosciences, San Jose, CA, USA, cat #550475) was used following the manufacturer’s protocol. Following trypsinization and two washes with cold PBS, 1 × 10^6^ cells were suspended in 1 mL of 1× binding buffer. Cells were then transferred into a new culture tube, and 5 µL of both APC Annexin V and 7-AAD were added to the solution along with 400 µL of 1× binding buffer. Samples were analyzed using an Accuri C6 cytometer (BD Biosciences, San Jose, CA, USA) using a flow rate of 35 µL/min, and 25,000 events were collected. Experimental controls consisted of an unstained sample, a sample treated with 5 µL of APC Annexin V only, and a sample treated with 5 µL of 7-AAD only. 

### 4.3. Invasion Chamber Assay 

Invasion chamber assays were performed using 50,000 cells from each cell line as previously described [16]. 

### 4.4. RNA Extraction and Real-Time PCR 

RNA extraction and qRT-PCR for gene expression were performed as described in [16]. All gene primers were obtained from Bio-Rad Laboratories (Mississauga, ON, Canada): *MXRA8* (qHsaCED0045992), *CDH1* (qMmuCID0005843), *HPRT* (qMmuCED0045738), *SNAI1* (qMmuCID0024342), *SNAI2* (qMmuCED0046072), *TWIST1* (qMmuCED0004065), *TWIST2* (qMmuCID0009652), *VIM* (qMmuCID0005527), *ZEB1* (qMmuCID0009095), and *ZEB2* (qMmuCID0014662). *HPRT* was used as the housekeeping gene.

### 4.5. RNA Sequencing

RNA sequencing was performed by Novogene Corporation Inc. (Sacramento, CA, USA) using total RNA extracted with the miRVana miRNA isolation kit (Thermo Fisher Scientific, Burlington, ON, Canada) as previously described [16]. Three independent samples were sequenced for the 231MXRA8Con, 231MXRA8KO-O, and 231MXRA8KO-V cell lines and mammary tumors induced by cell injections. Fastq files were processed using Genialis software v3.0 (Genialis Inc, Houston, TX, USA) following the standard RNA-seq pipeline which uses BBDuk to remove adapters and trim reads, STAR to align the reads, and featureCounts to generate gene-level counts. Hierarchical clustering and PCA analysis were performed using Genialis software v3.0 (Genialis Inc., Houston, TX, USA), and pathway analysis was performed using Enrichr software v3.0 [17,18]. The RNA sequencing data have been uploaded to the GEO database as accession number GSE238018.

### 4.6. Animals and Ethics 

Mice were housed and cared for following the guidelines established by the Central Animal Facility at the University of Guelph and the guidelines established by the Canadian Council of Animal Care. This study was approved by the Animal Care Committee at the University of Guelph (AUP #4838). 231MXRA8Con, 231MXRA8KO-V, and 231MXRA8KO-O cells were collected from logarithmically growing cultures, and 1 × 10^6^ cells were injected into the 4th mammary glands of female NCG (NOD-*Prkdc^em26Cd52^Il2rg^em26Cd22^*/NjuCrl) mice (Charles River, Wilmington MA, USA). Mice were monitored twice per week by palpating the mammary glands. Once a palpable mammary tumor was identified, the tumor size was measured using digital calipers. Initially, 3 mice were injected with each cell line, and the mammary tumors or mammary glands were collected when tumors induced by 231MXRA8Con cells reached approximately 10% of body weight (44 days post-injection). In a subsequent round of injections, seven mice were injected with each cell line, and mammary tumors were collected when the largest tumor reached approximately 10% of body weight (44 days post-injection for 231MXRA8Con tumors, 65 days post-injection for 231MXRA8KO-V tumors, and 90 days post-injection for 231MXRA8KO-O tumors). Mammary tumors were divided with a portion of the tumor being fixed in formalin for paraffin sectioning, embedded in OCT for frozen sectioning, or flash frozen for DNA, RNA, or protein analysis.

### 4.7. Tumor Specific Growth Rates (SGRs)

Tumor specific growth rates were calculated using the formula SGR = ln (V_2_/V_1_)/(t_2_ − t_1_), where V_1_ and V_2_ are the tumor volumes at the time of palpation (t_1_) and euthanasia (t_2_), respectively. 

### 4.8. Histology and Immunohistochemistry 

Mammary tumors and major organs were fixed in 10% formalin overnight and embedded in paraffin. Sections were cut and stained with hematoxylin and eosin for histologic analysis. Immunohistochemistry was performed as previously described [19], using a 1:200 dilution of an anti-vimentin antibody (ab16700, Abcam, Toronto, ON, Canada). Tissue sections were scanned using a Motic Easyscan digital slide scanner (Motic, Richmond, BC, Canada). Vimentin-stained sections were analyzed using the positive cell detection feature of the QuPath software v0.4.3 [20]. This software determines the percentage of vimentin-positive cells within the area of the image.

### 4.9. Tissue Microarray

The TMA BRM961a was purchased from ASMBIO (Cambridge, MA, USA). This TMA contained 17 ER^+^, 24 HER2^+^, and 8 triple-negative primary tumors and lymph node metastases from 11 ER^+^, 17 HER2^+^, and 8 TNBCs. Tumors were identified as ER^+^ if they were ER^+^/PR^+^/HER2^−^ or ER^+^/PR^−^/HER2^−^. The HER2^+^ tumors included any tumor that was HER2^+^ and thus tumors that were ER^−^/HER2^+^ or ER^+^/HER2^+^. All triple-negative tumors were ER^−^/PR^−^/HER2^−^. Immunohistochemistry of the TMA was performed with heat-mediated antigen retrieval with citrate buffer pH 6 then stained using a concentration of 1:100 for the anti-MXRA8 antibody (ab185444, Abcam, Toronto, ON, Canada). The TMA was scanned using a Motic Easyscan digital slide scanner (Motic, Richmond, BC, Canada) and analyzed with QuPath software v0.4.3 using the DAB channel only and the positive cell detection feature. The threshold was set to 0.2, and values above the threshold were considered positive, and values below the threshold were considered negative, with smoothing set to 1. The thresholding was restricted to regions of the core that contained tumor cells, and three independent regions were analyzed for each tumor, and then the average percentage of positive pixels was used and plotted.

### 4.10. Statistics 

To determine significance, an ANOVA test was performed followed by a Dunnett’s multiple comparisons test using GraphPad Prism 9.5 software (San Diego, CA, USA). Results were considered statistically significant when *p* < 0.05. 

## Figures and Tables

**Figure 1 ijms-24-13730-f001:**
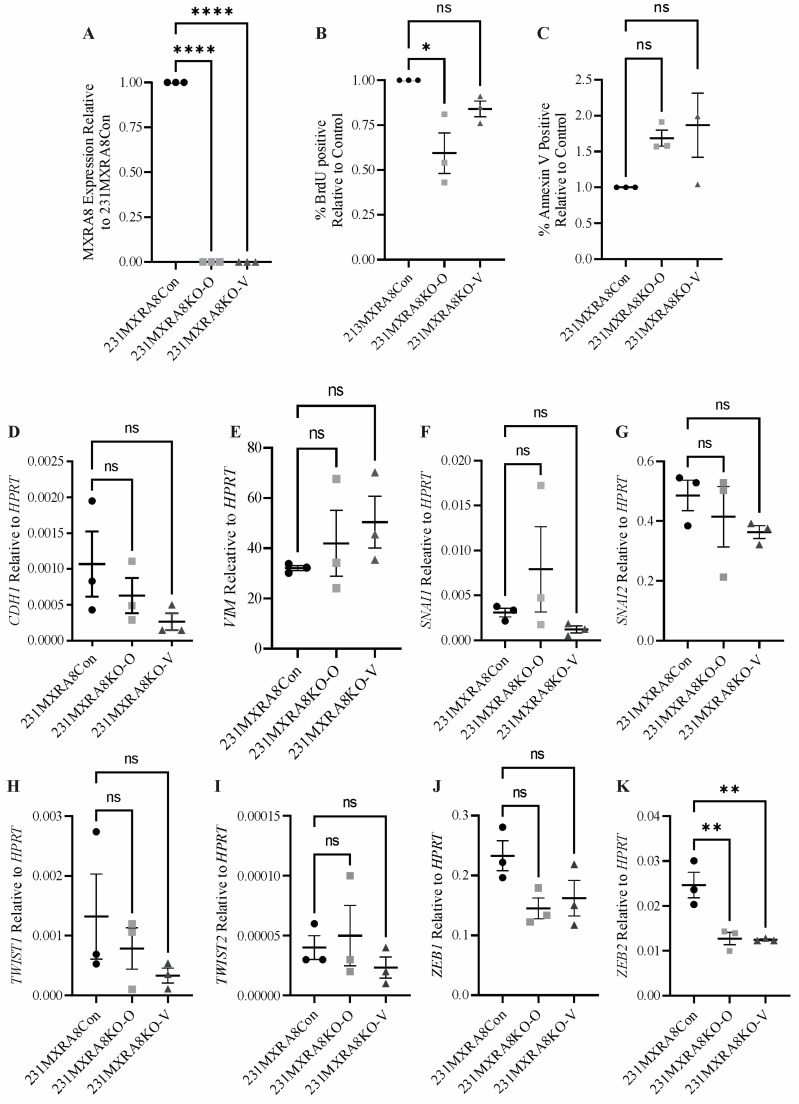
(**A**) The expression of *MXRA8* in control (231MXRA8Con) and *MXRA8*-knockout (231MXRA8KO-O and 231MXRA8KO-V) cells as assessed via qRT-PCR. The percentage of (**B**) BrdU-positive or (**C**) annexin V-positive cells in MXRA8-knockout clones relative to the control clone as determined using flow cytometry. The expression of (**D**) *CDH1*, (**E**) *VIM*, (**F**) *SNAI1*, (**G**) *SNAI2*, (**H**) *TWIST1*, (**I**) *TWIST2*, (**J**) *ZEB1*, and (**K**) *ZEB2* in 231MXRA8Con, 231MXRA8KO-O, and 231MXRA8KO-V cells as determined by qRT-PCR. Three individual samples were evaluated, and the horizontal line represents the mean of the samples. ns, non-significant, * *p* < 0.05, ** *p* < 0.01, **** *p* < 0.0001.

**Figure 2 ijms-24-13730-f002:**
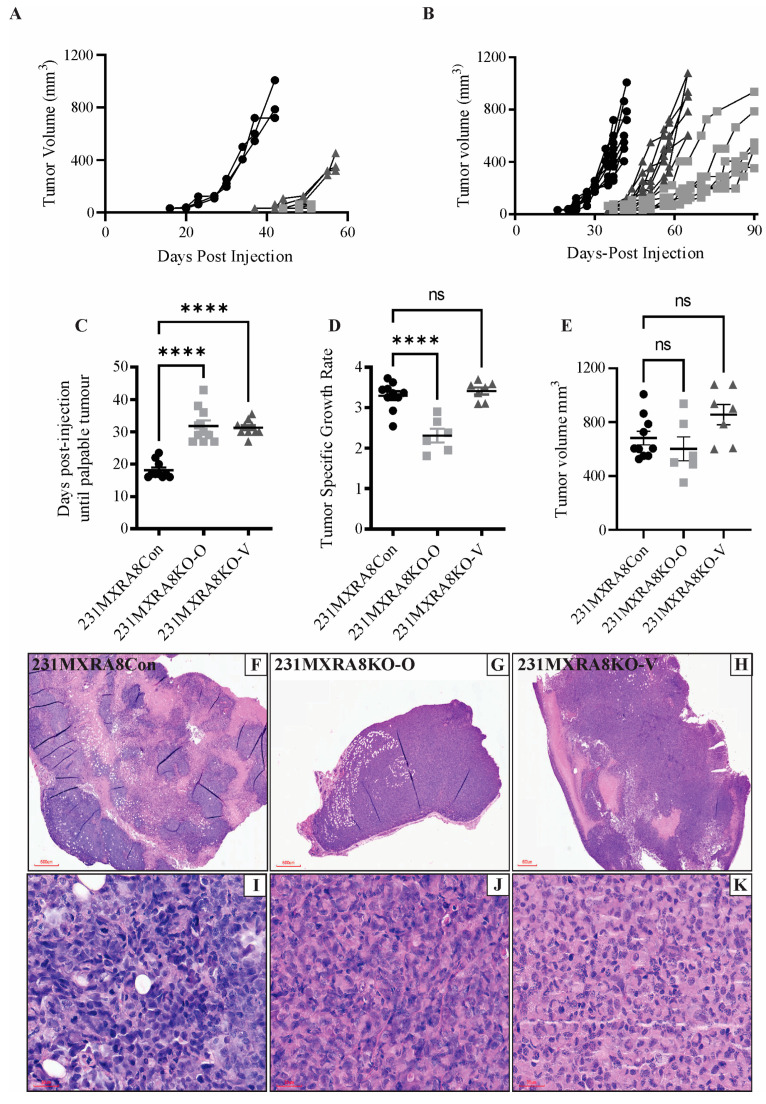
Tumor growth curves following the injection of 231MXRA8Con, 231MXRA8KO-O, and 231MXRA8KO-V cells when (**A**) tumors were collected once 231MXRA8Con tumors reached approximately 10% of body weight or (**B**) when the largest tumor from each cell line reached approximately 10% of body weight: (**C**) tumor onset, (**D**) tumor specific growth rate, and (**E**) tumor volume at the time of collections for tumors induced by injection of 231MXRA8Con, 231MXRA8KO-O, and 231MXRA8KO-V cells. Representative histological images of tumors induced by (**F**,**I**) 231MXRA8Con cells, (**G**,**J**) 231MXRA8KO-O cells, and (**H**,**K**) 231MXRA8KO-V cells. Scale bars for (**F**–**H**) are 600 μm, and scale bars for (**I**–**K**) are 30 μm. ns, non-significant, **** *p* < 0.0001.

**Figure 3 ijms-24-13730-f003:**
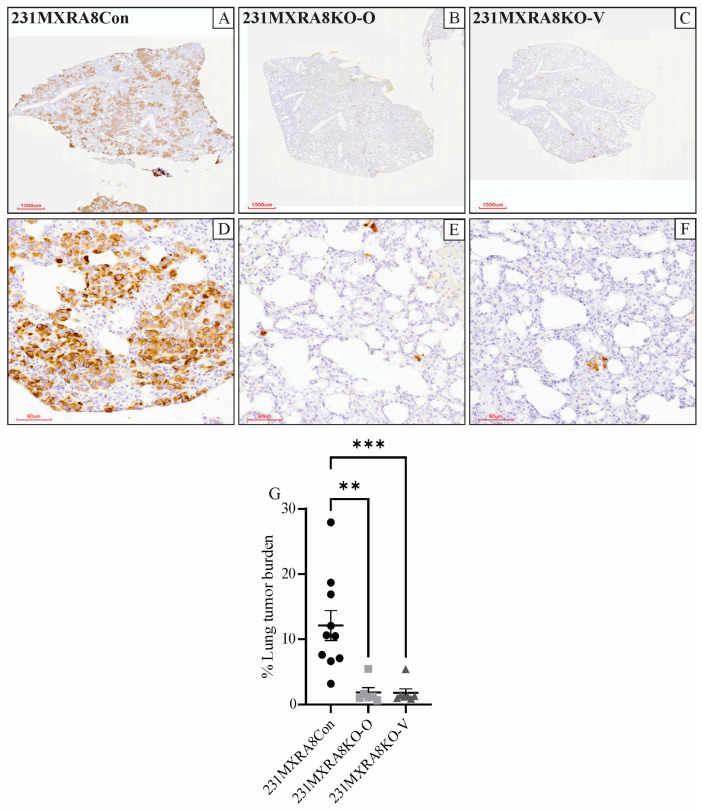
Lung sections following staining of human vimentin from mice harboring (**A**,**D**) 231MXRA8Con tumors, (**B**,**E**) 231MXRA8KO-O tumors, and (**C**,**F**) 231MXRA8KO-V tumors. Scale bars for (**A**–**C**) are 1000 μm and for (**D**–**F**) are 60 μm. (**G**) Quantification of the % of lung tissue containing vimentin-positive tumor cells. ** *p* < 0.01, *** *p* < 0.001.

**Figure 4 ijms-24-13730-f004:**
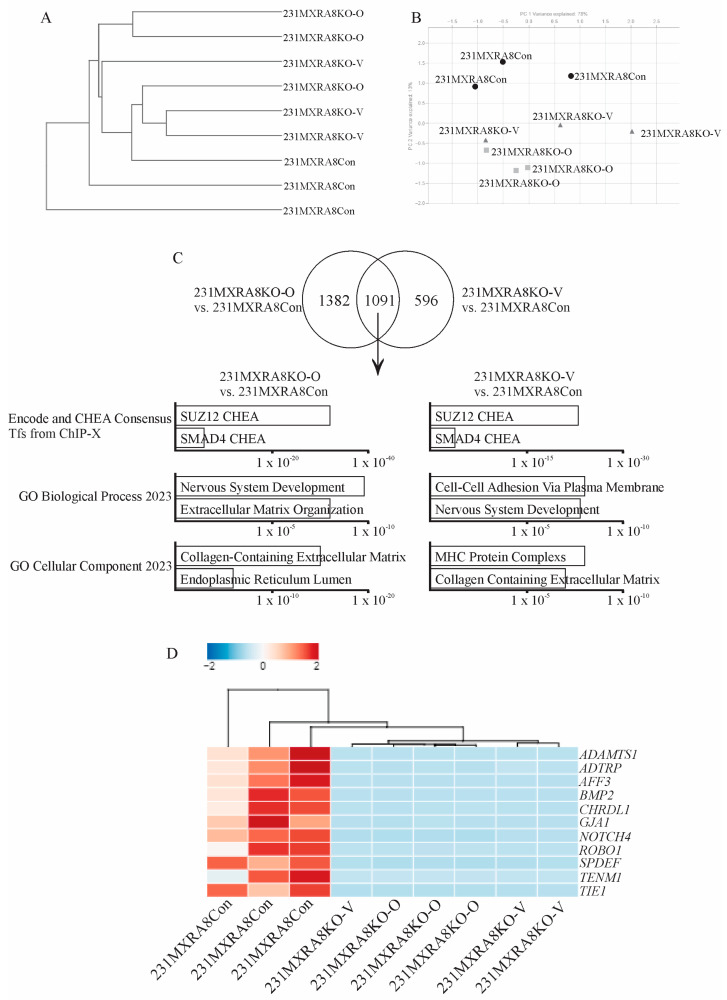
(**A**) Hierarchical clustering and (**B**) PCA analysis of the gene expression profiles of control and *MXRA8*-knockout tumors. (**C**) Venn diagram and pathway analysis of 231MXRA8KO-O tumors compared to 231MXRA8Con tumors and 231MXRA8KO-V tumors compared to 231MXRA8Con tumors. (**D**) Heatmap of the 11 genes found in the top-25 genes of both the 231MXRA8KO-O vs. 231MXRA8Con tumors and the 231MXRA8KO-V vs. 231MXRA8Con tumors based on the false discovery rate.

**Figure 5 ijms-24-13730-f005:**
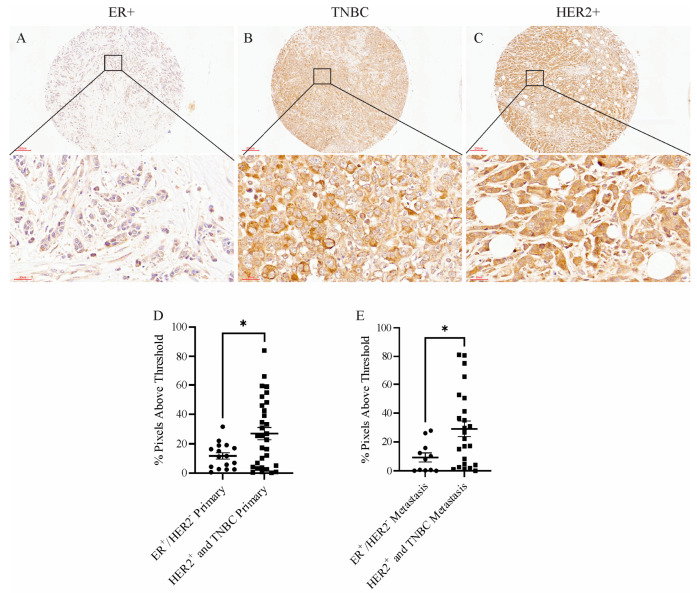
Representative images of MXRA8 staining of (**A**) an ER^+^ tumor, (**B**) a TNBC, and (**C**) a HER2^+^ tumor from a human breast cancer tissue array. Scale bars in the top images were 200 μm and in the bottom images were 30 μm. Quantification of the amount of MXRA8 staining in (**D**) primary tumors and (**E**) metastases from less aggressive breast cancers (ER^+^) compared to more aggressive breast cancers (TNBC and HER2^+^). * *p* < 0.05.

**Table 1 ijms-24-13730-t001:** TPM values for *MXRA8* protein-coding transcripts in *MXRA8* knockouts and control.

Transcript ID	Transcript Name	231MXRA8Con	231MXRA8KO-O	231MXRA8KO-V
ENST00000342753	*MXRA8*-202	1.8 ± 0.6	0.3 ± 0.2	0.1 ± 0.1
ENST00000309212	*MXRA8*-201	21.6 ± 2.1	0 ± 0	0 ± 0
ENST00000445648	*MXRA8*-203	3.8 ± 0.7	0 ± 0	0 ± 0

## Data Availability

The RNA sequencing data have been uploaded to GEO with the accession number GSE238018, and this data will be released as soon as the manuscript has been accepted for publication.

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
