# Peer review of "Loss of MXRA8 Delays Mammary Tumor Development and Impairs Metastasis"

_ijms, 2023, doi:10.3390/ijms241813730_

Round 1

Reviewer 1 Report

Thanks for the opportunity to review the paper titled, "Loss of MXRA8 Delays Mammary Tumor Development and Impairs Metastasis" by Kaitlyn E. Simpson et al. about the role of MXRA8 in regulating the progression of human triple negative breast cancer. This paper is interesting and might advance the scientific community. However, several issues should be addressed in this paper for publication in IJMS. Therefore, I consider this manuscript not suitable for publication. Nonetheless, I suggest to authors some feedback suggestions apply to other journals.

-        The abstract should be modified to give a brief description of the aim of this work, the methodology and the results. The introduction is not very clear. The connection between the novelty of the work and the intended purpose is not well-established, nor is it adequately linked to the existing background so far. The discussion is somewhat lacking. The limitations of your study are not included, nor do you discuss your results in comparison with current findings in lung cancer.

-        Lines 35-38. There is a contradiction in these two statements. While it is mentioned that the characterization in cancer is limited, it is also stated that in colorectal cancer, it has been associated with a poor prognosis.

-        The authors should explain why previous studies have been of a collective nature, their significance, and their distinct contribution in comparison to what has been studied thus far.

-        How does the reexpression of miR-200s in the TNBC cell line correlate to in vivo metastasis in breast tumors?

-        It should be clarified what the three cases in each group represent in the graphs of Figure 1. Why were a multiple comparisons analysis chosen, studying all the cases, instead of using the group mean? It is advisable to incorporate alternative analyses, as having only three cases per group is quite limited.

-        The baseline levels of apoptosis should be displayed in Figure 1.

-        Figures 2A and 2B lack a legend, making it unclear which group each line corresponds to.

-        In the histological image of Figure 2, there should be an outlined panoramic view displaying the entire tumor and non-tumor tissue, while the detail (2I-2J) should be presented at higher magnification to provide better visualization of the tumor cell morphology.

-        Why wasn't a tumor marker used to demonstrate that the cells in Figure 2 are indeed tumor cells? The authors should consider performing such an experiment.

-        In the legend of Figure 2, the letter 'L' is mentioned, yet there is no corresponding image labeled as 2L.

-        Why were the initial three injected mice not included in the second tumor cell injection, which involved 7 animals? Ethical committees typically strive to optimize the use of experimental animals.

-        How was the tumor-specific growth rate measured?

-        The scale bars for the images are missing in the legend of Figure 3. Additionally, images at a higher scale should be included.

-        In Figure 3, how has the percentage of lung tissue been quantified when even the cell nuclei are not visible? In my opinion, this approach is not accurate. It would be more appropriate to quantify the number of vimentin-positive cells per area.

-        How is the dual quantification in Figures 5D and 5E performed? From my understanding, I am concerned that this might not be the most suitable method. An alternative approach, such as dual immunofluorescence labeling with nuclear staining, could have been more appropriate. Could you explain the rationale behind conducting the microarray in this study? What valuable insights or contributions has the microarray analysis yielded?

-        Were the animals consistently sacrificed at 44, 65, and 90 days, respectively, according to the injected cell line? Did this always coincide with exactly 10% of their body weight?

-        The statistics section is incomplete; it does not mention how the assessment of normality was conducted, nor does it specify the post hoc test used.

Reviewer 2 Report

The manuscript titled “ Loss of MXRA8 delays mammary tumor development impairs metastasis” by Simpson et al., investigates the role of MXRA8 protein in delaying mammary tumor development and impairing metastasis. The reviewer has the following comments:

(i)             In line 45, author refers to their previous work, but do not provide a reference for the same. Citation needs to be updated.

(ii)           Authors need to provide a rationale for using single TNBC cell line MDA-MB-231. Why did they not use other TNBC cell lines or ER/PR positive cell lines.

(iii)         Characterization of MXRA8 knockout clone invitro: in Figure 1A, authors should expand the y-axis, right now, it is not clear what the difference in the relative expression of MXRA8 knockout clone (s) was in comparison to control. Secondly, did authors look at the MXRA8 protein expression levels in knockout clones and the controls?

(iv)          Was it a complete knockout or there was residual expression of MXRA8? Moreover, it is not indicated if the constitutive or inducible expression of MXRA8 was ablated in these cells?

(v)           For Figure 1B, authors need to expand the y-axis to clearly see the difference between knockout and the control

(vi)          What is the basal level of MXRA8 RNA and protein in MDA-MB-231 cell line?

(vii)        Repetition of the content from lines 104-133. Please delete the repeated content.

(viii)      Lines 144-147, Figure 2A and B tumors induced by control cells produced palpable tumors quickly compared to knockout clones-was this difference significant? Please elaborate when the authors refer to time as quickly?

(ix)          Did the authors observe any micro-metastases? If yes, in which tissues other than the lungs?

(x)           In Figure 5, is the MXRA8 protein expression cytoplasmic or nuclear? And please elaborate how the scoring was done to evaluate MXRA8 expression?

(xi)          Did authors evaluated by overexpressing MXRA8 gene to see if it actually promoted tumor development?
